# Is Vesicostomy Still a Contemporary Method of Managing Posterior Urethral Valves?

**DOI:** 10.3390/children9020138

**Published:** 2022-01-21

**Authors:** Aybike Hofmann, Maximilian Haider, Alexander Cox, Franziska Vauth, Wolfgang H. Rösch

**Affiliations:** 1Department of Pediatric Urology, Clinic St. Hedwig, University Medical Center Regensburg, 93049 Regensburg, Germany; alexander.cox@ukbonn.de (A.C.); franziska.vauth@barmherzige-regensburg.de (F.V.); wolfgang.roesch@barmherzige-regensburg.de (W.H.R.); 2Urologie Bayerwald, 94481 Grafenau, Germany; maximilian.haider@frg-kliniken.de; 3Department of Urology, University Hospital Bonn, 53127 Bonn, Germany

**Keywords:** posterior urethral valves, vesicostomy, congenital urinary tract obstruction, kidney function, vesicoureteral reflux, long-term outcome

## Abstract

In boys with posterior urethral valves (PUVs) the main treatment aim is to preserve long-term bladder and renal function. To determine the effectiveness of secondary vesicostomy in boys with PUVs, the medical records of 21 patients with PUV (2010–2019), divided into two groups (group I: valve ablation; group II: secondary vesicostomy), were reviewed regarding the course of serum creatinine, renal ultrasound, voiding cystourethrogram, urodynamics, postoperative complications, need of further surgery, and long-term solution. The median age of all patients at first follow-up was 11 (9–13) months and at last follow-up 64.5 (39.5–102.5) months. Despite a significant difference of the SWDR score (shape, wall, reflux, and diverticula) (*p* = 0.014), both groups showed no significant differences preoperatively. Postoperatively, serum creatinine (*p* = 0.024), grade of vesicoureteral reflux (*p* = 0.003), side of upper tract dilatation (*p* = 0.006), side of megaureter (*p* = 0.004), and SWDR score (*p* = 0.002) were significantly decreased in group II. Postoperative urodynamic measurements showed comparable results in both groups. Stoma complications were found in three (20%) patients (group II). Eight (53.3%) patients already received a closure of the vesicostomy. Seven out of eight (87.5%) patients were able to micturate spontaneously. Vesicostomy remains a reliable treatment option for boys with PUV to improve bladder function and avoid further damage to the urinary tract.

## 1. Introduction

Posterior urethral valves (PUVs) are the most common cause of lower urinary tract obstruction. PUVs were first described by Morgagni in 1717. Already in 1870, Tolmatschew recognized that PUVs are not an isolated anatomical defect [1] but a rather complex pathological entity that was described as ‘valve bladder syndrome’ by Mitchel [2]. This term emphasizes the interaction between altered bladder function, progressive hydronephrosis, and renal failure.

Recent decades have been marked by huge efforts, particularly in prenatal medicine, to optimize the treatment of male infants with PUV. Neonatal mortality rates have been decreased by early diagnosis, which was made possible by the widespread availability of prenatal ultrasound scans and intensive neonatal care [1]. However, long-term outcome is still poor because 30% of affected infants still develop chronic kidney disease before adolescence. Infants with PUV account for almost 17% of children with end-stage renal failure (ESRD) [3]. Despite its very low incidence that is estimated to range between 1:4000 and 1:7500, PUV remains a devastating disease [1].

Therefore, the main treatment aim is to relieve the pressure on and the obstruction of the urinary tract to improve bladder and renal function. Over the past few decades, urinary diversion has been replaced by initial catheter drainage followed by primary valve ablation [4,5]. Nevertheless, controversy remains regarding the most appropriate treatment in case of failure of early bladder decompression with urethral catheter drainage and primary valve ablation [1]. 

This study evaluated whether vesicostomy is still a striking tool in the treatment of PUV to preserve bladder and kidney function.

## 2. Materials and Methods

### 2.1. Patient Population

We identified all children with PUV who had been treated at our department (full member of ERN-eUROGEN) between January 2010 and June 2019. The population was divided into two groups. Group I included all patients after valve ablation solely and group II all patients after secondary vesicostomy, in whom valve ablation was performed before or during vesicostomy. Medical records were retrospectively reviewed in terms of the timing of valve ablation, the indication and timing of vesicostomy, the course of serum creatinine, results of renal ultrasound and voiding cystourethrogram (VCUG) before and (on average) one year after main surgery, urodynamic measurements after main surgery and before planned undiversion, free-flow after closure of the vesicostomy, surgical complications, febrile urinary tract infections (UTI), need of further surgery, long-term solution, and the total number of necessary operations. Exclusion criteria were previous operations affecting the bladder besides vesicostomy, late diagnosed posterior urethral valves, urethral atresia, other bladder diseases, and a follow-up period of less than 1 year.

### 2.2. Outcomes and Variables

The primary outcome was the course of serum creatinine as well as post-surgical alterations in the urinary tract after valve ablation, vesicostomy detected by means of sonographic or radiological examination and urodynamic measurements. Therefore, we analyzed serum creatinine values before surgery as well as the nadir that was defined as the lowest serum creatinine values in the year after valve ablation in group I and secondary vesicostomy in group II (hereafter referred to as main surgery). Ultrasound findings included the grade of hydronephrosis, alterations in renal parenchymal echogenicity, and differences in megaureter when present. VCUG findings were evaluated with regard to the grade of vesicoureteral reflux before and after main surgery and regarding radiographic changes in the bladder by means of the SWDR score (shape, wall, reflux, and diverticuli) determined by three assessors [6]. Kidneys were evaluated separately when appropriate. Urodynamic measurements were evaluated postoperatively in terms of bladder capacity and compliance as well as detrusor overactivity. Free flowmetries after undiversion were evaluated in terms of voided volume, percentage of age-related capacity, residual volume, and maximum urine flow. 

The secondary outcome included the requirement of further interventions because of persisting or recurrent UTI and complications due to vesicostomy. Additionally, the type of long-term solution was assessed when present.

### 2.3. Management and Follow-Up

All patients had received transurethral endoscopic diagnostics and, if applicable, primary, or repeated valve ablation with a cold knife. In the vesicostomy group, cutaneous vesicostomy was conducted using the Blocksom technique [7]. In both groups, all patients received low-dose antibiotics (trimethoprim 1 mg/kg body weight per day) for 4 weeks and concomitant anticholinergic therapy (oxybutynin 0.1 mg/kg body weight per day) after surgery. Postoperatively, follow-up was recommended every 3 months during the first year and every 6 months thereafter, alternating at the departments of pediatric nephrology and pediatric urology. On account of the large catchment area of our department, the follow-up examinations at 6, 9, and 12 months often took place close to the patients’ place of residence. At our department, each follow-up visit entailed a physical examination, evaluation of serum creatinine levels, as well as an ultrasound scan of the kidneys and the bladder. VCUG was obtained after a median period of 13 (range 4–54) months, and further VCUGs were conducted in case of clinical symptoms. Urodynamic measurements were obtained after a median period of 39 (range 11–71) months. Free flowmetries were obtained after a median period of 27 months (15–55.5) after closure of the vesicostomy. 

### 2.4. Statistical Analysis

Continuous variables are shown as median (interquartile range), median (range: minimum–maximum) or mean (±standard deviation) as appropriate. Categorical variables are expressed as counts and percentages. Comparisons of non-normally distributed continuous paired variables amongst the groups were performed with the Wilcoxon signed rank test and comparisons of categorical variables between groups with the Pearson’s x^2^ test or the Mann–Whitney U test as appropriate. All analyses were conducted using SPSS^®^, version 26.0 (IBM Corp., Armonk, NY, USA).

### 2.5. Ethics Statement

Ethical approval for this retrospective study was obtained from the Institutional Ethics Committee of the University of Regensburg (no. 20-2018-104).

## 3. Results

At our department, 40 patients had been treated for bladder valve syndrome between January 2010 and June 2019. The medical records of the patients were divided into two groups: Group I consisted of 11 patients after solely valve ablation; five patients had to be excluded because of late diagnosed PUV (one patient) and a follow-up of less than 1 year (four patients). Group II included 23 patients after secondary cutaneous vesicostomy according to the Blocksom technique; eight patients had to be excluded because of previous bladder surgery (two patients), urethral atresia (two patients), coexisting Ochoa syndrome (one patient), primary vesicostomy because of solely small caliber urethra (one patient), and a follow-up of less than 1 year (two patients). Consequently, the records of 21 patients were reviewed for this study. Valve ablation was carried out after a median period of 1 (0–2.5) month in both groups. In Group II, median age at urinary diversion was 2 (1–5) months. Median time between valve ablation and vesicostomy was 11 (0–64) days. The indications for secondary vesicostomy were functional single kidney and co-existing poor bladder function in 6 (40.0%) patients, persisting or increasing upper tract dilatation associated with increasing creatinine levels in four (26.7%) patients, and recurrent urinary tract infection in five (33.3%) patients. Overall median age at the first follow-up was 11 (9–13) months and at the last follow-up 64.5 (39.5–102.5) months. Detailed data are shown in Table 1.

Serum creatinine values were preoperatively the same in groups I and II. Although the median serum creatinine level remained postoperatively stable with 0.3 mg/dl in both groups, a significant decrease in preoperative and postoperative distribution was found in group II (*p* = 0.024). The creatinine nadir was achieved after a median period of 114 (40–316.5) days in all groups. Detailed data are shown in Table 2.

Preoperative ultrasound scans showed in terms of hydronephrosis no significant differences between the valve ablation group I and the vesicostomy group II. Although severe hydronephrosis was postoperatively decreased in all groups, the decrease was not significant compared to the preoperative values of either group. A significant (*p* = 0.004) effect was found in group II regarding the presence of a megaureter. In group I, one out of eight (16.6%) patients had already shown hyperechogenicity of the renal parenchyma in one kidney before surgery in contrast to 8/15 (53.3%) patients in group II, including one (3.3%) patient with a hypoplastic kidney and one (3.3%) patient with a non-visible kidney. 

Severity of the VUR grade showed a significant (*p* = 0.003) decrease after surgery, particularly in case of high-grade VUR in the vesicostomy group II. A comparison between the groups showed no significant difference between group I and group II (*p* = 0.231) The preoperative SWDR score differed significantly between groups I and group II (*p* = 0.014). Group I had a median preoperative score of 2 (range 1–5) and group II of 4 (range 3–6) up to a maximal score of 6. Group II showed a significant (*p* = 0.014) decrease in the SWDR score from a median score of 4 (range 3–6) to a median score of 1.5 (range 0–5). Pre- and postoperative images are shown in Figure 1.

Postoperative urodynamic measurements showed that normal bladder capacity could be achieved in three (75.0%) patients in group I and in eight (57.1%) patients in group II. In group I, two (50.3%) patients reached normal compliance in contrast to five (33.3%) patients in group II. Detrusor overactivity occurred in one (25.0%) patient in group I and in two (15.4%) in group II. Detailed data are shown in Table 3.

Urinary tract infection recurred in six (40%) patients in the vesicostomy group II and none (0%) in the valve ablation group. In terms of surgical outcome, stoma complications occurred in three (20%) patients in group II. Stoma revision was necessary in one (6.7%) patient in group II. Re-valve ablation was carried out in two (33.3%) patients in group I and in three (20.0%) patients in group II due to rest-valves. Detailed data are shown in Table 4.

Further surgery was necessary in two (33.3%) patients in group I and in three (20.0%) patients in group II before long-term solution. Detailed information about the surgical procedure is shown in Table 5.

Regarding the necessity of surgery: patients in group I required a median of 1.5 (range 1–3) surgical intervention, and patients in group II a median of 3 (range 3–6). Long term solution was achieved after a median of 58.5 (36.25–69.75) months. In group II, eight (53.3%) patients received an undiversion of the vesicostomy. In six (40.0%) patients, solely closure of vesicostomy could be performed. Two (13.4%) patients in group II and one (16.6%) patient in group I additionally needed a bladder augmentation. Seven (46.6%) patients required an additional antireflux surgery at the closure of vesicostomy. Bladder augmentation had to be performed in both patients in group II, due to bladder volume <50% of expected bladder capacity for age, functional single kidney, and persisting VUR, despite maximum conservative treatment. The patient in group I showed an increase of compliance with deterioration of the upper tract under maximum conservative therapy at the age of 5 years.

Seven (46.6%) patients in group II are still supplied with a vesicostomy. Detailed data are shown in Table 6.

In group II, seven out of eight (87.5%) patients are able to micturate spontaneously. The median percentage of the estimated maximum bladder capacity is 83.3% (58.3–100%). Detailed results of the free flowmetry and voiding parameters are shown in Table 7. 

## 4. Discussion

PUVs are one of the most serious types of congenital anomalies of the kidneys and the urinary tract. PUVs have a broad clinical spectrum and result in various dysfunctions of the urinary tract [8]. Although PUVs have been known for more than 100 years, the question which factors may improve the long-term outcome of renal and bladder function is still being controversially discussed [9,10]. Over the past decades, primary valve ablation has become the treatment of choice [4,5]. Despite the success of early valve ablation, the incidence of renal and bladder dysfunction varies widely [11]. Bladder dysfunction after valve ablation has been reported in up to 75% of male infants with PUV [12]. Hennus et al. stated in their review that there is still a lack of randomized studies showing that endoscopic treatment or retreatment may prevent the need for augmentation or ureteral reimplantation [9]. Due to the high rate of chronic kidney disease (CKD) and end-stage renal disease (ESRD), preserving renal function is one of the most important aims to be achieved in infants with PUV. In our study, the course of creatinine levels in the vesicostomy group (II) seemed to have been stable, with a median value of 0.3 mg/dl. However, we found a significant decrease in creatinine levels, particularly in initially high creatinine levels, after vesicostomy.

The prognostic value of nadir creatinine in the first year of life is well established. A nadir creatinine level of 0.8 mg/dL or less in the first year of life results in better long-term renal function [13,14]. In this regard, the various efforts to decrease high creatinine levels are well established. Any conditions that may increase creatinine levels at a later stage should be also considered. When comparing primary valve ablation and temporary vesicostomy with delayed valve ablation in boys with PUV, favorable outcomes of serum creatinine levels and estimated glomerular filtration rate (eGFR) in the vesicostomy group were published by Godbole et al. and Hosseini et al. [11,15]. Similar results were published by Prudente et al., who described decreased hydronephrosis and improved kidney function after vesicostomy [16]. Kim et al. described vesicostomy as a reliable surgical option in patients with severe PUV because this surgical method maintains renal function and guarantees adequate long-term bladder function [17]. On the other hand, Chua et al. found no long-term benefits for patients with CKD stage 3 but only temporarily delayed progression to ESRD. In their study, infants with PUV and CKD stage 3 were divided into three treatment groups: valve ablation, valve ablation plus subsequent vesicostomy, and valve ablation plus high urinary diversion. The three groups showed a statistically significant difference in progression to ESRD within 1 year but not at the 15-year follow-up. Besides postoperative eGFR, bilateral dysplasia seemed to be an independent variable for predicting CKD progression [18]. This finding may be related to the fact that high-grade CKD is more likely to be present in patients with renal dysplasia. As renal dysplasia is primarily a developmental malformation rather than an effect of obstructive uropathy [19], none of the described treatments would decrease renal impairment, which is caused by dysplastic kidneys. In consideration of this knowledge, we found progressing hyperechogenicity of the renal parenchyma after vesicostomy. We assumed dysplasia to be the underlying cause of this sonographic finding that cannot be influenced by any type of surgical treatment. In consideration of this fact, it seems much more important to stabilize and protect renal function of the contralateral non-dysplastic kidney as early as possible in cases with unilateral dysplasia. Any further obstruction due to persisting or newly developed VUR and hydronephrosis should be sufficiently removed as soon as possible to avoid recurrent UTI.

The present study showed a decrease especially of high grade VUR and hydronephrosis, as well as a reduction of recurrent UTIs in the vesicostomy group (II). 

In contrast, Bilgutay et al. showed in their analysis of 104 patients that the course of creatinine levels was the only independent risk factor for poor renal outcome [20]. Similar results were described by Çelakil et al. [10]. Despite the still ongoing discussion regarding the association of VUR and UTI with poor renal outcome, the current standard of knowledge is avoidance of both persisting VUR and recurrent UTI to preserve renal function [21]. Additionally, VUR and recurrent UTI are associated with multiple surgical interventions [20]. In our study, only a small number of infants required additional surgery because of VUR or recurrent UTI before closure of the vesicostomy. This finding may be attributable to a favorable effect of vesicostomy in patients with poor response to primary valve ablation.

In general, the role of urinary diversion in bladder function is being controversially discussed [11]. Opponents to urinary diversion describe poor bladder compliance, high detrusor pressure, and decreased bladder capacity. Additionally, patients with urinary diversion are assumed to require more surgical procedures and have a higher risk of bladder augmentation [4]. Although supravesical urinary diversion has fallen out of favor because it defunctionalizes the bladder, particularly in bilateral procedures, due to an arrest of physiological bladder cycling [1,4], the method seems still to commonly be used, not only in selected cases. Additionally, it has to be taken into account that all kinds of ureteral stoma formations compromise ureteral blood supply and may lead to lasting impairment of peristalsis, also, after ureteral reconstruction with or without ureteral reimplantation [22,23]. Astonishingly, some studies have shown a long-term benefit of defunctionalizing the bladder in terms of bladder compliance and function [1,24,25]. In 1974, Duckett was the first to describe cutaneous vesicostomy by means of the Blocksom technique as an alternative to primary valve ablation [7]. The position of the stoma allows bladder filling and preserves contractile function at reduced leak point pressure [26]. Whereas Puri et al. compared three groups of treatment in boys with PUV (fulguration, vesicostomy, and ureterostomy) and observed small-capacity bladders with hyperreflexia and hypertonia in the vesicostomy group [24]. Similar results were seen by Podesta et al.: patients with a vesicostomy had comparatively decreased bladder capacity and higher end-filling pressures compared to boys treated with valve ablation [3]. A significant disadvantage in toilet training, bladder compliance, and the necessity for augmentation in boys with diversion was postulated by Close et al. [5].

In our population, two (13.3%) patients in the vesicostomy group (II) required augmentation compared to one (12.5%) patient in the valve ablation group (I). Most of our patients already underwent closure of the vesicostomy and are able to micturate spontaneously with an adequate voiding volume for age.

With regard to the total number of necessary operations, patients in group II required a median of 2 more surgical interventions than patients in group I. Besides the creatinine values, the initial bladder situation in patients with PUV should also be taken into consideration. In addition to the sonographic and endoscopic findings, the SWDR score was used to objectify the bladder structure. Preoperatively, group I showed a significantly lower score than group II. In relation to postoperative outcomes, group II had a significantly decreased score. This result correlated well with our postoperative urodynamic findings, showing a comparable outcome in both groups.

Our study has several limitations. The retrospective design and the small sample size of the groups limits the generalization of our results. A proposed multicenter study could not be realized due to the center-specific different treatment pathways and the resulting difficulties in forming homogenous groups. CKD could not be accurately determined because of incomplete eGFR data. Despite these limitations, we were able to describe a highly homogenous group of patients, although PUVs are known as an anomaly with a wide range of clinical presentations and treatment options.

In conclusion, vesicostomy is beneficial in achieving renal function without the risk of damaging ureteral peristalsis and bladder dysfunction. It may even improve bladder function. Vesicostomy remains a reliable and safe treatment option for male infants with PUVs that may avoid high urinary diversion and prevent further damage to the upper and lower urinary tract.

## Figures and Tables

**Figure 1 children-09-00138-f001:**
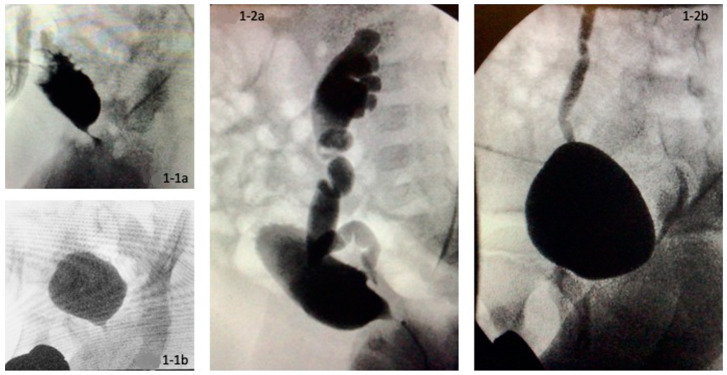
Pre- and postoperative voiding cystourethrograms of two patients (**1-1a**,**1-2a**). Preoperatively severely trabeculated bladder and vesicoureteral reflux (**1-1b**,**1-2b**). Normalization of bladder configuration and decrease in vesicoureteral reflux at 1-year follow-up.

**Table 1 children-09-00138-t001:** Patients’ demographics.

	Valve Ablation *n* = 6 (100.0%)	Secondary Vesicostomy *n* = 15 (100.0%)
Age at valve ablation in months (no. pat.)	6 (100.0%)	15 (100.0%)
Median (IQR)	1 (0.8–3.3)	1 (0–2)
Range	0–4	0–4
Age at urinary diversion in months (no. pat.)	*n*	15 (100.0%)
Median (IQR)		2 (1–5)
Range		0–16
Time between VA and urinary diversion in days (no. pat.)	*n*	15 (100.0%)
Median (IQR)		11 (0–64)
Range		0–427
Age at first follow-up in months (no. pat.)	6 (100.0%)	15 (100.0%)
Median (IQR)	11 (7.5–14.25)	11 (9–13)
Range	3–15	3–15
Age at last follow-up in months (no. pat.)	6 (100.0%)	15 (100%)
Median (IQR)	63 (46–95)	68.5 (34.75–105.25)
Range	37–173	3–118
Indication for urinary diversion (no. pat.)		15 (100.0%)
• Functional single kidney and poor bladder function	*n*	6 (40.0%)
• Abnormal renal function		4 (26.7%)
• Recurrent urinary tract infection		5 (33.3%)

**Table 2 children-09-00138-t002:** Initial and follow-up changes of laboratory values as well as sonographic and radiologic imaging.

	Valve Ablation *n* = 6 (100.0%)		Secondary Vesicostomy *n* = 15 (100.0%)		*p*-Value	
	pre-op	post-op	pre-op	post-op	pre-op	post-op
Serum Cr (mg/dl) (No. pat)	6 (100%)	6 (100%)	15 (100%)	15 (100%)	0.814	0.254
mean (±SD)	0.5 (±0.4)	0.3 (±0.5)	0.5 (±0.6)	0.3 (±0.1)		
median (IQR) range	0.3 (0.2–0.7) 0.2–1.2	0.3 (0.2–0.3) 0.2–0.3	0.3 (0.24–0.73) 0.2–2.4	0.3 (0.24–0.3) 0.2–0.6		
		*p = 0.225*		* **p = 0.024** *		
Side of upper tract dilatation						
No. patients	5 (83.0%)	6 (100.0%)	15 (100.0%)	15 (100.0%)		
None	0 (0%)	1 (16.7%)	0 (0%)	8 (53.3%)		
Unilateral	0 (0%)	1 (16.7%)	3 (20%)	2 (13.3%)		
Bilateral	5 (100%)	4 (66.7%)	12 (80%)	5 (33.3%)		
		*p = 0.180*		* **p = 0.006** *		
Grade of upper tract dilatation						
No. kidneys	11 (91.7%)	12 (100.0%)	29 (96.7%)	30 (100.0%)		
None	0 (0.0%)	4 (33.3%)	1 (3.4%)	10 (33.3%)	0.436	0.906
Mild (grade 1–2)	6 (54.5%)	5 (41.7%)	11 (37.9%)	13 (43.3%)		
Severe (grade 3–4)	5 (45.5%)	1 (8.3%)	15 (51.7%)	1 (3.3%)		
Hypoplastic kidney	0 (0.0%)	2 (16.7%)	1 (3.4%)	2 (6.7%)		
Dysplastic (non-visible)	0 (0.0%)	0 (0.0%)	1 (3.4%)	4 (13.3%)		
		*p = 0.313*		*p = 0.108*		
Side of Megaureter (>6 mm)						
No.pat.	4 (66.7%)	4 (66.7%)	14 (93.3%)	15 (100%)		
None	1 (25%)	2 (50%)	2 (14.3%)	4 (28.6%)	0.777	**0.012**
Unilateral	1 (25%)	2 (50%)	6 (42.9%)	5 (35.7%)		
Bilateral	2 (50%)	0 (0%)	6 (42.9%)	5 (35.7%)		
Side of vesicoureteral reflux		*p = 0.317*		* **p = 0.004** *		
No. patients	5 (83.3%)	6 (100%)	14 (93.3%)	14 (93.3%)		
None	2 (40%)	4 (66.7%)	4 (28.6%)	7 (50%)		
Unilateral	2 (40%)	1 (16.7%)	5 (35.7%)	4 (28.6%)		
Bilateral	1 (20%)	1 (16.7%)	5 (35.7%)	3 (21.4%)		
		*p = 0.317*		*p = 0.187*		
Grade of vesicoureteral reflux						
No. kidneys	11 (91.7%)	11 (91.7%)	27 (90.0%)	29 (96.7%)		
None	7 (63.6%)	6 (54.5%)	12 (44.4%)	22 (75.9%)	0.465	0.231
Mild (grade 1–2)	0 (0.0%)	1 (9.1%)	2 (7.4%)	3 (10.3%)		
Intermediate (grade 3)	0 (0.0%)	3 (27.3%)	2 (7.4%)	0 (0.0%)		
High (grade 4–5)	4 (36.4%)	1 (9.1%)	11 (40.7%)	4 (13.8%)		
		*p = 0.750*		* **p = 0.003** *		
SWDR score (No. pat.)	5 (83.3%)	6 (100%)	14 (93.3%)	14 (93.3%)		
median (minimum–maximum)	2 (1–5)	2 (1–6)	4 (3–6)	1.5 (0–5)	**0.014**	0.236
		*p = 0.317*		* **p = 0.002** *		

Pre- and postoperative *p*-values are marked in italics; significance (*p* < 0.05) is highlighted in bold.

**Table 3 children-09-00138-t003:** Postoperative outcome of urodynamic measurement.

	Valve Ablation *n* = 6 (100.0%)	Secondary Vesicostomy *n* = 15 (100.0%)	*p*-Value
Urodynamic filling parameters			
• Compliance (no. pat.)	4 (66.6%)	13 (86.6%)	
∘ Normal compliance	2 (50.0%)	5 (33.3%)	0.682
∘ Low compliance	2 (50.0%)	8 (53.3%)	
• DO * (no. pat.)	4 (66.6%)	13 (86.6%)	
∘ None DO	3 (75.0%)	11 (84.6%)	0.659
∘ With DO	1 (25.0%)	2 (15.4%)	
• Bladder Capacity (no. pat.)	4 (66.6%)	14 (93.3%)	
∘ Reduced capacity	0 (0.0%)	5 (35.7%)	0.289
∘ Normal capacity	3 (75.0%)	8 (57.1%)	
∘ Hyper capacity	1 (25.0%)	1 (7.1%)	

* DO = detrusor overactivity.

**Table 4 children-09-00138-t004:** Postoperative complications.

	Valve Ablation *n* = 6	Secondary Vesicostomy *n* = 15
Stoma complications (no. pat.)	*n*	3 (20.0%)
• Stoma prolapse		2 (13.3%)
• Stoma occlusion		1 (6.6%)
• Stoma revision		1 (6.6%)
Re-valve ablation (no. pat.)	2 (33.3%)	3 (20.0%)
Recurrent urinary tract infection (no. pat.)	0 (0%)	6 (40.0%)

**Table 5 children-09-00138-t005:** Case-related further surgery.

Pat. No.	Group *	Further Surgery
1	1	Re-valve ablation, bladderneck incision
2	1	Re-valve ablation, antireflux surgery
3	2	Conversion into ureterocutaneostomy, re-valve ablation
4	2	Revision of vesicostomy
5	2	Conversion into ureterocutaneostomy

* Group I: Valve ablation; Group II: Vesicostomy.

**Table 6 children-09-00138-t006:** Follow-up in terms of operative management.

	Valve Ablation *n* = 6	Secondary Vesicostomy *n* = 15
Long-term solution (no. pat.)		
• Closure of vesicostomy		6 (40.0%)
• Bladder augmentation without catheterizable stoma		1 (6.7%)
• Bladder augmentation and catheterizable stoma		1 (6.7%)
• Additional antireflux surgery	1 (16.6%)	7 (46.6%)
Age at long-term solution in years (no. pat.) Mean (±SD) Median (IQR)	1 (16.6%) 5 (±0) 5 (5–5)	7 (46.6%) 4.7 (±1.8) 5 (3–6)
Total number of operations Median (minimum–maximum)	1.5 (1–3)	3 (3–6)
Median (minimum–maximum)		

**Table 7 children-09-00138-t007:** Free flow/voiding results after undiversion.

Pat. No.	Age at Testing (Years)	Time After Undiversion (Months)	Procedure	Voided Volume	% Normal Max. Capacity	Max. Flow (mL/s)	Post-Void Residual Urine Vol. (mL)
1	11	47	Bladder augmentation without Stoma	260	0.72	7	100
2	6	15	Closure of vesicostomy	203	0.97	15.5	45
3	9	81	Closure of vesicostomy	350	1.16	10.8	30
4	8	28	Closure of vesicostomy	210	0.74	nn	0
5	4	15	Closure of vesicostomy		nn	nn	0
6	6	26	Closure of vesicostomy	252	1.2	18.3	50

## Data Availability

The data used to support the findings off this study are available from the corresponding author upon request.

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
