# Peer review of "Is Vesicostomy Still a Contemporary Method of Managing Posterior Urethral Valves?"

_children, 2022, doi:10.3390/children9020138_

Round 1
Reviewer 1 Report
In this study, the authors aimed to determine the effectiveness of vesicostomy in boys with posterior urethral valves (PUV). They compared the long-term clinical outcomes of three patient groups undergoing valve ablation, vesicostomy and ureterocutaneostomy due to PUV.
There is a major problem with the design of the study, it is not clear from the manuscript if how many patients eventually underwent vesicostomy reversal and when were the imaging studies made (at the time of un-diversion, after some time, before)? The patient status is extremely important to assess the outcomes. There is a sentence in the discussion stating ‘Most of our patients underwent closure of the vesicostomy and were able to micturate spontaneously’. What do the authors mean? They havent given this knowledge in the results. Are there patients with permanent vesicostomy and how do the authors plan to manage them?
Furthermore, if the authors wanted to give the outcome of vesicostomies why did they include a cutaneous urethrostomy group? It really doesn’t make sense since the 3rd group including higher diversions are completely different even at the presentation. I would suggest to take that group out or change the title. Below you can find my further remarks.
- When showing non-normally distributed variables, I recommend “median (IQR)” instead of minimum-maximum values. In addition, some results suggesting that it was calculated incorrectly were given. For example, "Age at urinary diversion in months was shown as 4.67 ± 5.8". The fact that the standard deviation is larger than the mean in this expression indicates that there may have been a statistical error. If there is a non-parametric distribution, it may be necessary to change the statistical test. Also, mean creatinine levels of the 2nd group practically did not change but the p value seems to be <0.05. I understand the distribution issue but it just doesn’t fit in. I think it would be useful to get help from a specialist in the field of statistical analysis.
- There are 8 patients in valve ablation, 15 patients in vesicostomy, and 6 patients in cutaneous urethrostomy. This patient population is too small to perform an ideal statistical analysis. This limitation greatly reduces the power of the study. Accordingly, the paper does not have the data information required to specifically analyze the stated situation.
- What is main surgery? Is it valve ablation for group 1, vesicostomy for the group 2 and cutaneous ureterostomy for group3 or valve ablation for all, or else? Please clarify the fate of all patients to give a clear perspective to the reader.
- Please give more information on the rates of rest ablation/urethral strictures?
- We already expect reduction in tresolution of hydronephrosis after vesicostomy since the bladder pressures are leveld to athmosperic pressure. And those with valve ablation only are undergoing previously described phases of valve-bladder.
- Cutaneous urethrostomy and UTIs does not fit, are they clinically confirmed febrile UTIs with high fever and labaratory findings?
- Describe ‘shriveled kidney’ in the text, since it is not a common terminology. Hypoplasia may fit better if it is meant to be.
- Augmentation should be deferred as much as possible in PUV patients, as their bladder dynamics change over time. It should be better stated why augmentation cystoplasty was considered.
It should be kept in mind that although initially similar in some parameters, those who required vesicostomy in the follow-up are more hostile bladders.
Bottomline, is there a role for vesicostomy in PUV? Definitely, there is however the paper doesn’t come up with a different knowledge on any related issue.
P.S I would suggest the authors to bring out the SWRD score forward as there are very few studies mentioning it.
Author Response
Dear Reviewer 1,
we highly appreciate the chance to further work on our manuscript in order to have it published in Children.
Your comments proved to be helpful and constructive, and we feel able to respond accordingly to each of them. Please find a point-by-point response, including information on the changes made to the manuscript below.
Comments are in italic, changes to the manuscript have been implemented by using the ´Track Change`function.
We would like to express our gratitude for your fair and thoughtful review, raising important points yet underestimated in our manuscript.
Reviewer #1,
In this study, the authors aimed to determine the effectiveness of vesicostomy in boys with posterior urethral valves (PUV). They compared the long-term clinical outcomes of three patient groups undergoing valve ablation, vesicostomy and ureterocutaneostomy due to PUV.
There is a major problem with the design of the study, it is not clear from the manuscript if how many patients eventually underwent vesicostomy reversal and when were the imaging studies made (at the time of un-diversion, after some time, before)? The patient status is extremely important to assess the outcomes. There is a sentence in the discussion stating ‘Most of our patients underwent closure of the vesicostomy and were able to micturate spontaneously’. What do the authors mean? They havent given this knowledge in the results. Are there patients with permanent vesicostomy and how do the authors plan to manage them?
We added the number of patients, who already underwent closure of the vesicostomy, and the number of patients, which are still conducted with a vesicostomy. Ultrasound and VCUG were in average performed in the first year after surgery. Urodynamic studies were performed before undiversion was planned. None of the patient is planned for a permanent vesicostomy.
We added the missing information to the manuscript.
Furthermore, if the authors wanted to give the outcome of vesicostomies why did they include a cutaneous urethrostomy group? It really doesn’t make sense since the 3rd group including higher diversions are completely different even at the presentation. I would suggest to take that group out or change the title. Below you can find my further remarks.
The major aim of our study was to show the effect of the vesicostomy especially with regard to long-term bladder function, therefore we compared the three treatment options. We were aware of the different presentation of the third group, especially with regard to the renal impairment. But both groups (vesicostomy and ureterostomy) showed initially a comparable severeness of bladder dysfunction in the radiographic studies. On the other hand, we know that the severeness of renal damage occurs already in early gestation age, independent from the bladder development. So, we wanted to focus on two major points, which can be affected by our therapy:1.) bladder development in depending on our treatment option and 2.) prevention of deterioration of the upper tract, regardless of the initial presentation.
But we agree also with your objections, so that we decided to re-design our study and take the third group out.
- When showing non-normally distributed variables, I recommend “median (IQR)” instead of minimum-maximum values. In addition, some results suggesting that it was calculated incorrectly were given. For example, "Age at urinary diversion in months was shown as 4.67 ± 5.8". The fact that the standard deviation is larger than the mean in this expression indicates that there may have been a statistical error. If there is a non-parametric distribution, it may be necessary to change the statistical test. Also, mean creatinine levels of the 2nd group practically did not change but the p value seems to be <0.05. I understand the distribution issue but it just doesn’t fit in. I think it would be useful to get help from a specialist in the field of statistical analysis.
For the revision the statistician of our university was consulted for the statistical analysis. The statistical test were corrected. The statistical test for the creatinine level was again perfomed and is according to our statisicia correct.
- There are 8 patients in valve ablation, 15 patients in vesicostomy, and 6 patients in cutaneous urethrostomy. This patient population is too small to perform an ideal statistical analysis. This limitation greatly reduces the power of the study. Accordingly, the paper does not have the data information required to specifically analyze the stated situation.
The small number of patients is indeed a limitation of our study. But we excluded rigorously all patients with additional treatment to perform a homogenous group as possible for each treatment in this very widely presented disease.
- What is main surgery? Is it valve ablation for group 1, vesicostomy for the group 2 and cutaneous ureterostomy for group3 or valve ablation for all, or else? Please clarify the fate of all patients to give a clear perspective to the reader.
Main surgery is valve ablation solely in group I, secondary vesicostomy after valve ablation in group 2, and it was urtereostomy after valve ablation for group 3. We clarified this in the manuscript.
- Please give more information on the rates of rest ablation/urethral strictures?
We add the information to the manuscript.
- We already expect reduction in tresolution of hydronephrosis after vesicostomy since the bladder pressures are leveld to athmosperic pressure. And those with valve ablation only are undergoing previously described phases of valve-bladder.
- Cutaneous urethrostomy and UTIs does not fit, are they clinically confirmed febrile UTIs with high fever and labaratory findings?
Yes, this have been clinically confirmed UTIS, with catheter-urine collected out of the ureterostomy.
- Describe ‘shriveled kidney’ in the text, since it is not a common terminology. Hypoplasia may fit better if it is meant to be.
We changed the terminology into hyoplastic kidney
- Augmentation should be deferred as much as possible in PUV patients, as their bladder dynamics change over time. It should be better stated why augmentation cystoplasty was considered.
In our institution in boys with PUV augmentation is only performed in very selected cases, as we are aware of the changes of the bladder due to myogenic failure in adolescence.
The reasons for augmentation were added in the manuscript.
It should be kept in mind that although initially similar in some parameters, those who required vesicostomy in the follow-up are more hostile bladders.
Yes, we fully agree
Bottomline, is there a role for vesicostomy in PUV? Definitely, there is however the paper doesn’t come up with a different knowledge on any related issue.
In our daily work we recognized that the vesicostomy is increasingly being critically regarded and performed less in case of failure of decompression of the bladder after valve ablation in boys with PUV. Additionally, some papers assume a worsening of bladder function, if vesicostomy is performed. Therefore, we evaluated our population in terms of effectiveness of the vesicostomy.
P.S I would suggest the authors to bring out the SWRD score forward as there are very few studies mentioning it.
This is a really interesting issue, we are planning a study comparing urodynamics with VCUG to bring the SWRD score forward
Reviewer 2 Report
Dear authors,
This is a very interesting manuscript. However, I would have a few suggestions.
- The introduction to the Abstract is too abrupt. I recommend an introductory phrase.
- The abbreviation SWRD must be explained at the first appearance, in Abstract and the main text (shape, wall, reflux and diverticuli); the same for DO (Detrusor overactivity)
- Please clarify this: in the Methods section, first paragraph : "Group I consisted of 11 patients after solely valve ablation", while in the third paragraph: "Median age at urinary diversion in group I was 2 (range 0-19) months".
- In Table 1, first row, Age at valve ablation: What happened to the 29th patient?
- The data in Table 2 have been disturbed in word processing and are difficult to follow
- Please review the matching of references in the text; for example, in the Discussion section: "Hennus et al. stated in their review that there is still a lack of randomized studies showing that endoscopic treatment or re-treatment may prevent the need for augmentation or ureteral implantation [12]." Your reference no. 12 is "PARKHOUSE, H.F.; Barratt, T.; Dillon, M.; Duffy, P.; Fay, J.; Ransley, P.; Woodhouse, C.; Williams, D. Longterm outcome of boys with posterior urethral valves. British journal of urology 1988, 62, 59-62."
- The very small sample size is a major limitation of this study. Have you considered a multicenter study?
- The Statement about Data Availability must be removed/adjusted
- The references must be standardized
Author Response
Reviewer #2:
Comments and Suggestions for Authors
Dear authors,
This is a very interesting manuscript. However, I would have a few suggestions.
- The introduction to the Abstract is too abrupt. I recommend an introductory phrase.
We changed the introductory phrase
- The abbreviation SWRD must be explained at the first appearance, in Abstract and the main text (shape, wall, reflux and diverticuli); the same for DO (Detrusor overactivity)
Both abbreviations are now explained at the first appearance
- Please clarify this: in the Methods section, first paragraph : "Group I consisted of 11 patients after solely valve ablation", while in the third paragraph: "Median age at urinary diversion in group I was 2 (range 0-19) months".
The first paragraph is corrected
- In Table 1, first row, Age at valve ablation: What happened to the 29th patient?
Due to the change of the study design, we are now able to provide the age of valve ablation of all patients.
- The data in Table 2 have been disturbed in word processing and are difficult to follow
We improved all tables, to make them easier to follow
- Please review the matching of references in the text; for example, in the Discussion section: "Hennus et al. stated in their review that there is still a lack of randomized studies showing that endoscopic treatment or re-treatment may prevent the need for augmentation or ureteral implantation [12]." Your reference no. 12 is "PARKHOUSE, H.F.; Barratt, T.; Dillon, M.; Duffy, P.; Fay, J.; Ransley, P.; Woodhouse, C.; Williams, D. Longterm outcome of boys with posterior urethral valves. British journal of urology 1988, 62, 59-62."
We reviewed the references and corrected them
- The very small sample size is a major limitation of this study. Have you considered a multicenter study?
Yes, we have considered a multicenter study, but due to the often different therapy pathways it is very difficult to perform homogenous groups in this varying disease. Additionally, we recognized that not many centers are performing a secondary vesicostomy anymore. A fact, which also led us to the decision to publish our data.
- The Statement about Data Availability must be removed/adjusted
The Statement about Data Availability has been removed
- The references must be standardized
The references were edited with the bibliography program EndNote 19 and the reference style “mdpi” as required from the journal
Round 2
Reviewer 1 Report
Thank you for the revision.
Author Response
Thank you very much for the acceptance of our revision.
Reviewer 2 Report
Dear authors,
Due to the change of your study design, now it is a completely new study. I am not sure it is better that way, considering the even smaller number of patients included and compared, but the results are certainly clearer. Anyway, VUP is a rare, complicated disease, and there is no standardized managemnet; I think your work is important.
- Please verify the last phrase in the abstract: "Vesicostomy remains a reliable treatment option for boys with PUV to prevent bladder function and further damage to the urinary tract."
- Please include your previous answer in the manuscript, when you discuss about the limitation of the study: "Yes, we have considered a multicenter study, but due to the often different therapy pathways it is very difficult to perform homogenous groups in this varying disease. Additionally, we recognized that not many centers are performing a secondary vesicostomy anymore. A fact, which also led us to the decision to publish our data."
- You excluded the group III, so please reformulate the following phrase from the discussion section: "This result correlated well with our postoperative urody-namic findings, showing a significantly higher rate of normal bladder capacity in group II than in group III and comparable rates in relation to bladder compliance and detrusor overactivity in the group I."
Author Response
Dear Reviewer,
thank you very much for your further revision.
Please verify the last phrase in the abstract: "Vesicostomy remains a reliable treatment option for boys with PUV to prevent bladder function and further damage to the urinary tract."
We corrected the last sentence in the abstract into: "Vesicostomy remains a reliable treatment option for boys with PUV to improve bladder function and avoid further damage to the urinary tract."
Please include your previous answer in the manuscript, when you discuss about the limitation of the study: "Yes, we have considered a multicenter study, but due to the often different therapy pathways it is very difficult to perform homogenous groups in this varying disease. Additionally, we recognized that not many centers are performing a secondary vesicostomy anymore. A fact, which also led us to the decision to publish our data."
We included the following sentence in the manuscript: "A proposed multicenter study could not be realized, due to the center specific different treatment pathways and the resulting difficulties to perform homogenous groups."
You excluded the group III, so please reformulate the following phrase from the discussion section: "This result correlated well with our postoperative urody-namic findings, showing a significantly higher rate of normal bladder capacity in group II than in group III and comparable rates in relation to bladder compliance and detrusor overactivity in the group I."
We corrected the phrase in the discussion section.